# The Influence of Risk Factors in the Early Detection of Developmental Dysplasia of the Hip in a Country with Limited Material Resources

**DOI:** 10.3390/healthcare11172416

**Published:** 2023-08-29

**Authors:** Iuliana-Laura Candussi, Daniela Ene, Camelia Bușilă, Raul Mihailov, Ioan Sârbu, Claudiu N. Lungu, Carmen Iulia Ciongradi

**Affiliations:** 1Department of Pediatric and Orthopaedic Surgery, Clinical County Children Emergency Hospital, “Dunarea de Jos” University, 800010 Galati, Romania; iuliana.candussi@ugal.ro (I.-L.C.);; 2Department of Pediatrics, Clinical County Children Emergency Hospital, “Dunarea de Jos” University, 800010 Galati, Romania; camelia_busila@yahoo.com; 3Department of Surgery, Faculty of Medicine and Pharmacy, “Dunarea de Jos” University, 800010 Galati, Romania; raul.mihailov@ugal.ro; 42nd Department of Surgery—Pediatric Surgery and Orthopedics, “Grigore T. Popa” University of Medicine and Pharmacy, 700115 Iași, Romania; carmen.ciongradi@umfiasi.ro; 5Department of Surgery, Clinical County Emergency Hospital, 810325 Braila, Romania

**Keywords:** DDH, risk factors, early detection, children, hip ultrasound

## Abstract

Developmental dysplasia of the hip (DDH) is a condition that includes a wide spectrum of anomalies ranging from simple instability with ligamentous hyperlaxity to the complete displacement of the femoral head outside the abnormally developed cotyloid cavity. Early detection and initiation of treatment allow “restitutio ad integrum” healing, which has raised the medical community’s interest in early diagnosis. However, in countries with limited material resources, where echographic screening is not performed, efforts are being made to increase the sensitivity of clinical screening. Thus, the concept of “hip at risk” is taking shape worldwide. This is the normal clinical hip, but associated with one or more risk factors. We conducted a retrospective study for the period 2010–2015 with patients who presented in the ambulatory clinic of the St. John Children’s Clinical Hospital, Galati. The study included 560 patients, who were all examined clinically and sonographically, according to the Graf method, by a senior orthopedic doctor with competence in hip sonography. The data obtained from the anamnesis, clinical examination, and ultrasound examination were recorded in the DDH file. The goal of the statistical analysis of the group of patients was to find a correlation between DDH and the risk factors used in the clinical detection of this pathology. In the studied group, four risk factors were identified that have an increased association with DDH: female sex, pelvic presentation, limitation of coxo-femoral abduction, and congenital clubfoot; thus, the conclusion of the study is that patients who have at least one of the listed risk factors should be examined sonographically as quickly as possible. The early ultrasound examination will allow the identification of the disease and the initiation of treatment.

## 1. Introduction

Developmental dysplasia of the hip (DDH) is a condition that includes a wide spectrum of anomalies ranging from simple instability with ligamentous hyperlaxity to the complete displacement of the femoral head outside the abnormally developed cotyloid cavity. DDH is a pathology with evolutive potential. The structures of the coxo-femoral joint that are normal during embryogenesis become pathological. The American Academy of Pediatrics defines DDH as a pathology in which the femoral head no longer maintains normal anatomical relationships with the acetabulum.

DDH is a child pathology that has been of interest and controversy in the medical community since the time of Hippocrates. Today, even with early detection through clinical and imaging screening methods, this injury remains a real challenge for the orthopedic surgeon.

Preventive medicine is taking shape internationally, with clinical applications in all medical specialties. Health can no longer be conceived as a gift but as the result of daily concerns on the part of each individual and the society of which they are a part. Therefore, it is necessary to convince the population of the value and profitability of prevention, rather than only orienting medical thinking and medical-sanitary actions prophylactically. Some societies have come to a conclusion, based on studies, that prevention, rather than treatment, brings material, moral, social, and psychological benefits to patients [1,2]. Thallinger and collaborators highlighted in their paper presented in 2014 the long-term results of national screening for DDH [3,4].

The “hip at risk” concept emerges from attempting to complete a clinical picture. This is a normal hip on clinical examination but is likely to become dysplastic [5]. Currently, several factors are well recognized as fetal and maternal risk factors. However, although they increase the risk of DDH, the majority (73–90%) of patients diagnosed with DDH do not have risk factors, as suggested by Chan and Gharedaghi in their works [6,7]. Therefore, they must also be interpreted in a clinical context, representing a way to guide the diagnosis. The data from the specialized literature are vast and confusing, due to different definitions, different diagnostic methods (clinical, radiological, ultrasound, or MRI examination), different ages of the examined populations (newborn, 1 month, 3 months, 6 months) and studies conducted on different populations. Currently, the diagnosis and treatment of DDH is carried out depending on the clinical experience of the examiner, without the possibility of quantifying the lesion and fitting it into a treatment algorithm.

Due to the plasticity of the two components of the coxo-femoral joint, detected and treated early, DDH present at birth can be cured “restitutio ad integrum” [8]. Currently, in Romania, the diagnosis of DDH is made at the age of 6 months based on a radiograph of the pelvis. Early diagnosis, based on hip ultrasound, represents an insufficiently documented area in therapeutic terms, but because there is no possibility of performing a screening, it is necessary to perform a selective screening based only on the presence of risk factors and a clinical examination. The clinical examination of the newborn provides few clinical elements for the detection of DDH, but the presence of one or more risk factors in a patient with a negative clinical examination can direct the patient to perform a hip ultrasound, thus allowing the early diagnosis of DDH.

The association of several factors with the risk of DDH occurrence is recognized in the specialized literature. These factors, described by Dimeglio, are sex, pelvic presentation, primiparity, family history, high birth weight, prematurity/postmaturity, disorders of muscle tone, congenital disorders of the foot, abduction limitation, postnatal position, torticollis, twinship, geographical area and ethnicity, oligohydramnios, pathology side [9,10].

The incidence of hip dysplasia in Romania is currently still unknown [10]. Due to a low socio-economic level, in Romania, ultrasound screening for DDH using the Graf method is only possible in a few medical centers. The neonatologist and the family doctor carry out a routine clinical examination, but often the clinical examination is inconclusive. The extent of the problem can be known if the incidence of DDH is known. The only reference in the literature to the incidence of DDH in Romania is of 1% in northwest Transylvania, but the year of study is not specified. International studies have highlighted an increased incidence of DDH in Caucasians from Eastern Europe, so we can conclude that our country is in an endemic area for DDH. In recent years, the medical community’s interest in DDH screening has been observed, but an official reporting of the cases has yet to be observed. The purpose of the study is to find a statistically relevant association between risk factors and the presence of DDH.

## 2. Materials and Methods

The study we carried out is a retrospective one carried out regarding the period from 2010–2015. The patients included in the report presented themselves in the ambulatory service of the Children’s Emergency Clinical Hospital “St. John” Galati. Each patient was clinically examined, a hip ultrasound was performed, and a DDH sheet was later drawn up.

In the period 2010–2015, 634 patients presented themselves. All were examined clinically and sonographically, but 74 were excluded. Of these, 32 were older than 6 months, 20 had neurological conditions, 1 had arthrogryposis, and 21 had an incomplete DDH file (Figure 1).

Following the exclusion of patients with neurological diseases, with neurological damage of central origin, and those whose age at the first orthopedic examination exceeded 6 months, 560 patients were included in the study (Table 1).

In the study, the patients were examined clinically and sonographically by a senior doctor with experience in this pathology and competence in hip sonography according to the Graf method. During the inspection, data regarding the asymmetry of the adductor folds, the asymmetry of the gluteal folds, the inequality of the limbs, the obliquity of the vulvar slit, signs of the ascent of the greater trochanter, and other signs evocative of DDH were observed and noted.

Through the bimanual examination technique, the Ortolani and Barlow maneuvers were performed.

By positioning the knee and the hip in 90-degree flexion, the abduction amplitude was measured with the goniometer. It was considered that the value of 75 degrees is the limit between normal and pathological.

The technical support with which the ultrasound was performed was a Mindray 50 ultrasound machine, equipped with iDMS (intelligent data management system) software, with the possibility of the cranial–caudal orientation of the image and adjustment of the ultrasound penetration distance (useful in macrosomic patients or those with sizeable adipose tissue), automatic angle measurement software, and Graf classification. The system is equipped with a 7.5 Mhz linear probe, an image freeze pedal, and a Mitsubishi thermal printer.

To perform an ultrasound examination under optimal conditions, a device for lateral hip examination is necessary. The cradle from the original Graf technique is called Sonofix and consists of a polyurethane frame lined with a sponge.

This system allows the immobilization of the baby in the desired position and access to the examined balance. To perform the ultrasound correctly, we used a cradle that was adjusted to the original specifications of the product.

The ultrasound probe guide system, Sonoguide, limits its movements and obtains images only in the standard anterior–posterior plane. Performing ultrasound without a Sonoguide predisposes results to the phenomenon of “tilting” [11], leading to errors in evaluating the reference points necessary for drawing the baseline.

The data obtained from the anamnesis, clinical examination, and hip ultrasound were used to prepare a DDH file. To this were added the type of treatment and its duration; if a bilateral coxo-femoral radiograph was performed with the patellae at the zenith and with abduction and internal rotation; the residual acetabular dysplasia; the acetabular index; and the 15 risk factors of DDH mentioned above.



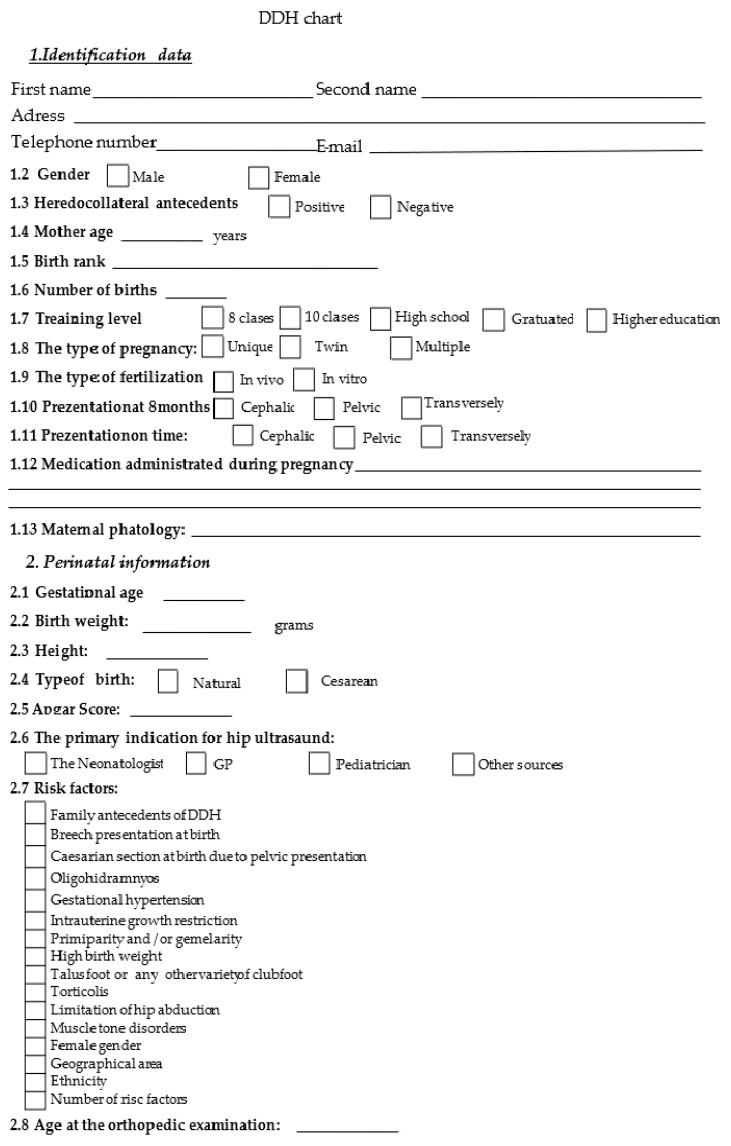


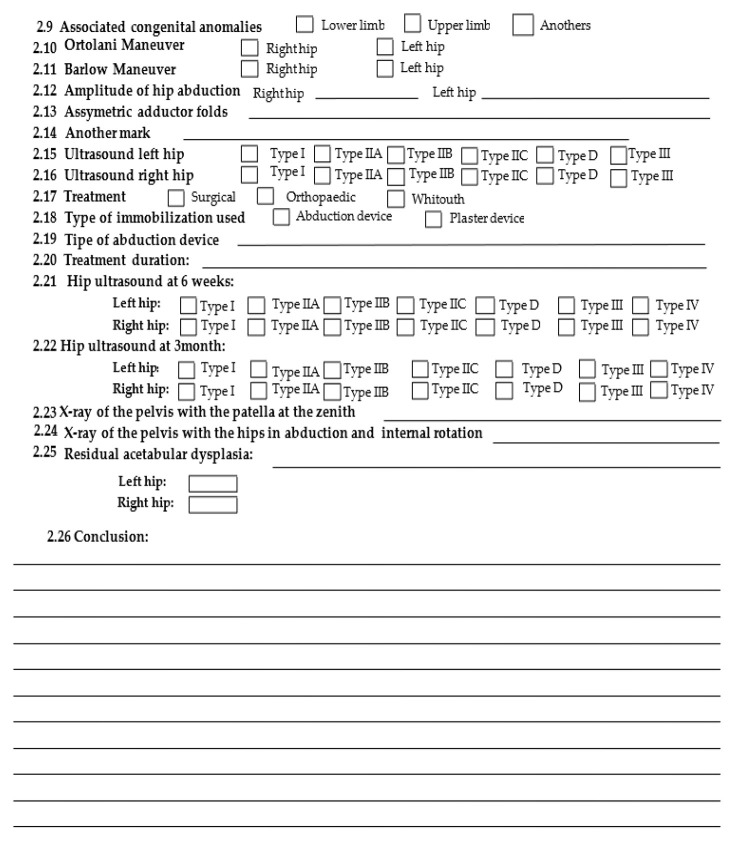



For data collection, based on the DDH assessment sheet, we created a database in Microsoft Access for entering and storing information electronically. Later, the data were processed in Microsoft Excel 2016 and IBM SPSS version 23.

For the existing variables, we applied descriptive statistics methods to highlight the characteristics of the studied plots. For the purpose of comparison between the various analyzed groups, we used percentage values.

In order to test the working hypotheses, we applied specific statistical tests depending on the type of data analyzed. To consider the result a statistically significant test, we used the threshold value to accept the alternative hypothesis, *p* ≤ 0.05. Also, where this was the case, we used the 95% confidence interval to estimate the results.

Among the statistical tests used are: the Levene test to determine the homogeneity of variances, the t-test for comparisons between groups, the Shapiro–Wilk test to determine whether the analyzed distributions differ statistically significantly from a normal distribution or not, the non-parametric Mann–Whitney U test for the cases in which the criteria for the application of the t-test were not met, the Chi-square test for verifying the association between two variables, and where the conditions for its application were not met, the Fisher test or the likelihood test, depending on the type of data analyzed and the odds ratio for risk determination.

## 3. Results

With the statistical analysis of the group of patients, we tried to find a correlation between DDH and the risk factors used in the clinical detection of this pathology. Five hundred and sixty patients were clinically and sonographically examined between January 2010 and December 2015. Of the 560 patients, 486 (87%) were clinically and sonographically unchanged, and 74 (13%) presented with clinical or sonographic signs of a dysplastic hip (Table 2).

Of the 560 patients, 399 (71%) were female, and 161 (29%) were male.

In the group of 560 patients, 57 were in pelvic complete, incomplete, and frank presentation, and 503 were in other types of presentation. Applying the Chi-square test revealed a *p*-value of 0.0224, which is statistically significant and demonstrates the causal relationship between pelvic presentation and DDH. The majority of patients came from an urban environment, 448 patients (80%), in contrast to those from a rural environment, 112 (20%).

We used the Chi-square test to observe if there is a statistically significant relationship between primiparity and the occurrence of DDH; *p* is 0.1135, so it is statistically insignificant. Thus, statistically, in our study, this element does not influence the occurrence of DDH. However, the large number of patients from a first pregnancy who applied for consultation reveals the parents’ prudence and vigilance.

Of the total of 560 patients, 62% of mothers were college graduates, and 33% were high school graduates; a minimal number had only primary school, at 2%, or ten classes, at 3%. Oligohydramnios in the patients of the studied group was not present. Therefore, due to insufficient data, we consider that the analysis is statistically insignificant.

In the group of patients studied, no significant differences were found between DDH and high birth weight; the *p*-value was 0.6394.

In the presented group of 560 patients, 10 had muscular torticollis. We applied the Chi-square test to establish a relationship between the two pathological entities, and the test result was statistically insignificant (*p* has a value of 0.9584).

In the studied group, 88 patients had a variety of congenital foot disorders, of which six also presented DDH. The Chi-square test found a statistically significant difference in the association of the two entities, the *p*-value being 0.0004. From the total of 560 patients, only 14 presented abduction limitation, of which seven patients showed it to be due to DDH. The applied Chi-square test revealed the existence of a statistical correlation between the two elements (*p* = 0.0042). The study results correspond to the statistics from the literature: Castelein [12] and Gulati [13].

## 4. Discussion

Since the development of hip ultrasound as a method for early detection of DDH, there has been engagement in the medical community. As a non-invasive method, parents received it well, and the demand for this investigation increased significantly. This investigation’s technical support is not complex; it can serve several medical specialties (radiology, neonatology, pediatrics, orthopedics, and family medicine). However, sometimes not all steps required to perform a hip ultrasound are performed correctly, causing inconclusive results. If all the conditions and stages of ultrasound are followed, as described in the Graf technique, it is a safe and easy test. Still, if performed incorrectly, it can produce false-negative results with disastrous consequences for the patient’s health [5].

The concept of “hip at risk” is emerging from attempts to complete a clinical picture. This is a normal hip on clinical examination, but likely to become dysplastic. Currently, several factors are well recognized as fetal and maternal risk factors. However, although they increase the risk of DDH, the majority (73–90%) of patients diagnosed with DDH do not have risk factors [6,7].

1.Sex—DDH represents a pathology that is more common in girls, the rate being 4/1 or 6/1 [8]. This is due to the sensitivity of the female fetus to maternal hormones (especially relaxin), which causes ligamentous laxity. It is known that the female sex is favored in the occurrence of this injury. The examined group shows a difference in the proportion of female patients diagnosed with DDH. The Chi-square test highlighted a cause–effect relationship between the female sex and the presence of DDH; the *p*-value is 0.00133, which is statistically significant. This statistic concludes that the female sex is associated with an increased risk of DDH and that the addressability is higher in girls. DDH etiopathogenesis and risk factors were reviewed by Harsanyi et al., who noticed that female newborns are nine times more frequently diagnosed with DDH than male newborns [11].2.Pelvic presentation—Pelvic presentation is known to increase the risk of DDH. In a complete pelvic presentation, the thighs are flexed on the abdomen, and due to the pressure of the uterus and abdominal wall muscles, the femoral head is forced posteriorly. In addition, the permanent flexion of the thighs leads to the shortening of the iliopsoas, which is an important factor in postnatal dislocation. In the incomplete pelvic presentation, the instability of the hip increases due to the increased tension in the hamstrings due to the position of the knees in extension. Studies have shown an increased incidence in patients with pelvic presentation born vaginally (17.5%) compared to those born by cesarean section (10.3%) [9]. The statistical results of our study are consistent with the data from the literature. Hsieh, in his study [14], states that he believes that DDH is part of the complex of injuries secondary to the pelvic presentation, including torticollis and facial anomalies. This statistically proves that a patient in a pelvic presentation will have a risk of 2.337 (statistically significant) to develop DDH compared to a newborn in another type of presentation.3.Primiparity—Due to the particulars of the primiparous pregnant woman, the first newborn has a higher incidence: lack of elasticity of the uterus, increased tonicity of the abdominal muscles, and oligohydramnios.4.The location of the injury—DDH is more frequent on the left because the fetus is positioned with its back to the left side of the mother, and the thigh is placed on the sacrum. It is positioned in flexion and adduction, increasing the risk of dysplasia. In addition, the right hip is more exposed in the pelvic presentation [15].5.Family history—Studies have shown that compared to dizygotic twins, monozygotic twins have a higher incidence, and positive family history for DDH increases the risk of the pathology by 10%.6.High birth weight—A weight over 4000–4500 g increases the risk of DDH.7.Prematurity/postmaturity—The risk of DDH seems lower in premature infants than in postmature ones, supporting the theory that DDH appears late in intrauterine life.8.Disorders of muscle tone—The slow progression of the femoral head in the socket may result from early muscle hypertonia.9.Congenital disorders of the foot—Studies have shown that metatarsus adductus and talus valgus foot associate in a proportion of 20% with DDH, although initially it was considered that equine varus foot was associated with DDH. In the group presented, a congenital foot disorder was represented by different entities: talus valgus foot, metatarsus adductus, and equine varus foot. Both rigid and flexible forms represent these forms. Although the number of patients with DDH and congenital foot disorders is not large, the association is statistically significant between the two pathologies is in the literature’s data. The common pathogenic substrate of the two pathologies is ligament hyperlaxity; congenital foot disorders represent its evident clinical expression, and DDH represents its hidden variant.10.Abduction limitation—Abduction limitation represents a critical risk factor in detecting DDH. It is also a conclusive clinical sign. Still, in newborns, the limitation of abduction can be secondary to physiological muscle hypertonia. Limitation of abduction is a valuable clinical sign; it suggests, in most cases, a problem at the hip level, and unilateral limitation has increased specificity for DDH. Bilateral limitation of abduction, especially when not significant, can be erroneously interpreted by the examining physician as a deviation from normal. The statistical risk that a patient with abduction limitation will present DDH is 22.9% (CI 95% 4.51–16.19). It is a high risk from a statistical point of view.11.Postnatal position—A still-used practice is swaddling the newborn by fixing the lower limbs in extension and adduction. However, in this position, all the conditions for developing DDH are created through maximum coxo-femoral instability and the association with ligament hyperlaxity. In 1975, according to a study in Japan, the incidence of DDH was reduced by half by avoiding swaddling the newborn with maximum knee extension and adduction.12.Oligohydramnios—Oligohydramnios in the patients of the studied group was not present. Therefore, due to insufficient data, we consider that the analysis is statistically insignificant.13.Twinship—Twinship is a risk factor that supports the mechanical theory of DDH occurrence. Due to the inadequate intrauterine space, twinning or multiple pregnancies were considered for a long time as an etiopathogenic factor in the occurrence of DDH, but the studies performed did not highlight a significant connection between the two elements [12,16].14.Geographical area and ethnicity—International studies have highlighted an increased incidence of DDH in Caucasians from Eastern Europe, so we can conclude that our country is in an endemic area for DDH.15.Torticollis—Torticollis is associated with DDH in 15% of cases.

Although they are not known as risk factors, the environment of origin and the degree of training of the parents have an important determinism in the identification of DDH.

We believe that the large proportion of patients from the urban environment is due to their easy access to information, the existence of a better addressability of the family doctor, and a better level of preparation of the parents.

Of the total of 560 patients, 62% of mothers were college graduates, and 33% were high school graduates; a minimal number finished only primary school (2%) or ten classes (3%). The mother’s training level cannot influence the etiology of DDH. However, an educated mother will not ignore the appearance of a new clinical element and will show up for another consultation from a specialized physician. She will present herself to the specialist doctor for re-examination, speeding up the diagnosis and treatment process. Thus, based on the current batch, the training level can become an essential factor in detecting DDH, and the natural evolution of the disease can be improved by vigilance over the patients. Under these conditions, we can say that DDH is a “**disease of ignorance**”.

In 1975, Czeizel and collaborators established, based on a study, that the first-degree relatives of a person with DDH have an eight times higher risk of DDH than the general population, and the risk to second-degree relatives is four times higher than the general population [17,18]. Today, the genetic determinism of the pathology is known, but the mode of transmission and its clinical expression are unknown, and in our research, the value p-0.5759 did not highlight a statistically significant correlation between family history and DDH.

Although, in Romania, the medical community considers this method effective and helpful in diagnosing DDH, there is no specialized training center for medical personnel to detect this pathology. For this reason, such investigations are often carried out by inadequate personnel, without a solid knowledge of this pathology, and under inadequate technical conditions. This often leads to false-positive or false-negative results, leading to treatment errors and decreased optimism about the capabilities of this early detection method. In Romania, the first clinical examination of the coxo-femoral joint in newborns is usually performed by a neonatologist, followed by a family doctor.

Increasing the sensitivity of the clinical screening is achieved by dynamically performing the clinical examination. However, this method has some disadvantages:It must be performed by the same medical staff;It congests the outpatient service;It can delay diagnosis and treatment.

In countries where ultrasound screening is impossible, finding some signs/symptoms/elements or their associations that raise the suspicion of hip dysplasia is necessary [19,20].

Thus, the results of our study showed, through statistical data from a correct evaluation carried out by an excellently trained staff, that the presence of one of the four factors—female sex, pelvic presentation, congenital clubfoot, limitation of abduction—increases the risk of DDH.

The introduction of hip ultrasound as an early detection method initially determined the increase in the incidence of DDH by detecting and including the physiologically immature hip in the dysplastic hip category. This was why it was established that the optimal age for performing the ultrasound is 4–6 weeks. Thus, the incidence does not represent an indicator in the assessment of the early detection of DDH.

But the effectiveness of clinical screening has been discredited over time, due to the increased percentage of false-negative results. Due to the multiple clinical forms (dysplastic, subluxated, luxating, unstable hip), clinical screening is not only ineffective but also risky.

Although this form of detection has been applied for 50 years by the world medical community, the rate of complications and costs in treating DDH have not decreased [21].

In the examined group, 89% had a negative clinical picture at the time of the examination, but of these, 12 patients had a dysplastic hip during the ultrasound examination. In medical practice, the real challenge is to detect a dysplastic hip, in a timely manner, from the group of those with a negative clinical examination.

The other studied factors had different statistical values, but without significant values from this point of view. However, other retrospective studies have highlighted significant values for different factors. These differences appear due to the heterogeneity of the studied groups and socio-cultural differences.

Harsanyi et al. reviewed DDH etiopathogenesis and risk factors and remarked the following: in the literature, the female gender shows seven to nine times more frequent DDH diagnoses at birth than male newborns [14]. Isolated right hip dysplasia is the least common type. The most common is the affection of the left hip due to fetal positions where the left hip is leaning towards the spine of the mother. Bilateral affection is also frequent [22]. Vaginal delivery, although having other benefits compared to cesarean section, significantly increases the statistical risk for DDH [23,24]. Factors such as oligohydramnios, high birth weight (H.B.W.), or primiparity [25] present an increased risk for DDH [26,27]. Specific populations have reported higher incidence rates of DDH due to tight swaddling techniques [18,28].

Ibrahim et al. conducted a retrospective study of DDH risk factors in Saudi Arabia. The study was conducted using the medical records of 82 children born or admitted to the King Abdul Aziz Medical Center in Riyadh, Saudi Arabia, with clinically suspected hip dislocation (DDH). The correlation between DDH and the following risk factors was investigated: age < 3 years, female sex, parental consanguinity, firstborn, cesarean section, pelvic presentation, prematurity, positive family history, and the presence of associated anomalies. The authors concluded that positive family history, female gender, age < 3 years, and the presence of associated abnormalities indicated an approximately 16, 3, 2.5, and 2-fold increased risk for DDH [20].

Finally, Kural et al., after examining a total of 19,516 hips from 9758 children for DDH, found that 57 had 97 hips with abnormal ultrasonographic findings [29]. The female gender was found to have a significantly high prevalence among the children in the case group. Limited hip abduction and positive Ortolani and Barlow signs were vital clinical findings in the case group. The study concluded that breech presentation, female gender, torticollis, and multiple pregnancies were risk factors for this disorder. Infants with these risk factors could be carefully screened for DDH.

In the study we conducted, we identified four major risk factors that are associated with DDH: female gender, pelvic presentation, congenital disorders of the foot, and limitation of abduction. However, the study we carried out has limitations: it does not present a control group, the sample of patients is small, there is no national registry to refer to, and the post hoc power is 84.7.

## 5. Conclusions

As a result of our study, we have grouped the risk factors into two categories: major risk factors (pelvic presentation, female sex, limitation of abduction, congenital clubfoot) and minor risk factors (oligohydramnios, torticollis, high birth weight, ethnicity, primiparity, postnatal position, prematurity/postmaturity, disorders of muscle tone, twinship). The ultrasound examination must be performed as quickly as possible in patients with a major risk factor, and within 4–6 weeks for the rest of the patients, thus optimizing the treatment and shortening the treatment period. The number of risk factors does not influence the risk of developing hip dysplasia compared to each single factor. Developmental dysplasia of the hip is a disease that can be cured by “restitutio ad integrum,” constituting a pathology that can be included in screening programs, and hip ultrasound is the method of choice for early detection.

## Figures and Tables

**Figure 1 healthcare-11-02416-f001:**
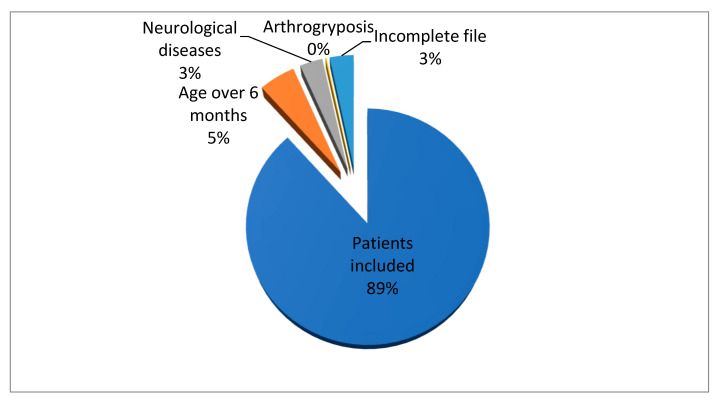
Case distribution in the studied group.

**Table 1 healthcare-11-02416-t001:** Inclusion and exclusion criteria.

Inclusion Criteria	Exclusion Criteria
Age ≤ 6 months	Age ≥ 6 months
First presentation	Arthrogryposis
Complete DDH chart	Incomplete DDH chart
	Neurological disorders

**Table 2 healthcare-11-02416-t002:** DDH risk factors.

Risk Factors	*p*	Chi
**Sex**	**0.0133**	**19.31**
**Pelvic presentation**	**0.0224**	**17.85**
Primiparity	0.1153	12.90
Location of injury	0.9925	1.51
Family history	0.5759	6.64
High birth weight	0.6394	6.07
Prematurity/Postmaturity	0.9434	2.85
Disorders of muscle tone	0.8471	4.11
**Congenital disorders of the foot**	**0.0004**	**28.53**
**Limitation of hip abduction**	**0.0042**	**22.4**
Postnatal position	0.9512	2.72
Oligohydramnios	0.9434	2.85
Twinship	0.1646	11.71
Torticollis	0.9584	2.57
Geographical area and ethnicity	0.9246	3.15

## Data Availability

Data are available on reasonable demand.

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
