# Peer review of "The Influence of Risk Factors in the Early Detection of Developmental Dysplasia of the Hip in a Country with Limited Material Resources"

_healthcare, 2023, doi:10.3390/healthcare11172416_

Round 1

Reviewer 1 Report

Please see the attachment file.

The quality of writing in English is not appropriate.

Reviewer 2 Report

The study attempted to identify risk factors for DDH and produced meaningful results.

The need for early ultrasound examination for patients with high risk factors is emphasised, which seems to be correct, but this does not seem to be explicitly indicated in the results and statistical analysis.

This may be off the subject of this paper, but I think that seasonal variations in occurrence (more common in winter) were known, and it will be meaningful to include information whether there were significant differences also in this study, if possible.

It must be careful in statement from line 238 to 242, since the statistically insignificant does not mean no influence of primiparity on DDH but might only indicates the sample is not large enough to show statistical significance.  

As minor points,

In table 1:

Age>6months -> Age ≧ 6months

 Artrogripoza -> Arthrogryposis

in line 149: The period is missing at the end of the sentence.

Reviewer 3 Report

- In the introduction, maybe elaborate on all the listed risk factors (ie oligohydramnios and geographic area) are listed but nothing else is stated. What is known and why it impacts risk? (if known)

- Last paragraph of the methods section is not clear. Recommend revising. 

Line 114: missed the "H" in D.D.
Line 126: Delete "Several"
Line 149: missed the period at the end of the sentence
Line 212: add "with" to state "...presented WITH clinical or sonographic..."
Line 230: Should say "This statistically proves that..."?
Line 243-244: Needs to be reworded
Line 300: Remove the "comma" after In Romania.
Line 317: Comma should be a dash--> "... limitation of abduction- increase..."
Line 350-356: These are 2 run on sentences whose punctuation do not make sense. Recommend adjusting. 

General comments:
-Maintain correct tense. Are you speaking in past or present tense?
-Watch for extra spaces between words and punctuation

Round 2

Reviewer 1 Report

None.

Author Response

I have attached the file with the answers.

Round 3

Reviewer 1 Report

Thank you for your effort to respond to the reviewer's comments. Unfortunately, the introduction and discussion sections have not been dramatically improved. The detected risk factors in this study are well known risk factors, and new factors were not found. The evidence added to the current body of literatures by this study is trivial. Thus, the publication of this study may not be possible considering the quality of study design and writing.

None.
